# Physiological Aspects of Milk Somatic Cell Count in Small Ruminants—A Review

**Shehadeh Kaskous** [1,*] **, Sabine Farschtschi** [2] **and Michael W. Pfaffl** [2]

[1] Department of Research and Development, Siliconform, Schelmengriesstrasse 1, 86842 Türkheim, Germany
[2] Department of Animal Physiology and Immunology, TUM School of Life Sciences, Technical University of Munich, Weihenstephan, Weihenstephaner Berg 3, 85354 Freising-Weihenstephan, Germany
* Correspondence: skaskous@siliconform.com

**Abstract:** The aim of this review was to focus on the physiological aspects of milk somatic cell count (SCC) in small ruminants (SM). The SCC is an important component naturally present in milk and is generally used as an indicator of milk quality and udder health in milk producing ruminants. SCC contains the following cells: polymorphonuclear neutrophils (PMN), macrophages, lymphocytes, and many milk epithelial (MEC) cells, cell fragments, and cytoplasmic particles/vesicles. PMN (40–80%) represent the major cell type in milk in healthy uninfected goats, whereas the macrophages (45–88%) are the major cell type in sheep's milk. However, dairy goats and sheep have an apocrine secretory system that produces cytoplasmic cellular particles/vesicles and large numbers of cell fragments, resulting in the physiological SCC limit being exceeded. It is obvious that the SCC level in milk of SM can be affected by various influencing factors, such as milk fraction, breed, stage of lactation, parity, type of birth, milking system, and others. An increase in the SCC above the physiological level not only indicates an udder or general health problem but reduces milk production, changes the milk composition, and hence affects milk processing. Moreover, the milking machine plays an important role in maintaining udder health in SM and stable SCC at physiological levels in the milk obtained. So far, there are no healthy or pathological physiological SCC levels defined in SM milk. Furthermore, a differential cell count (DCC) or even a high resolution DCC (HRDCC), which were recently developed for cattle milk, could also help in SM to gain deeper insight into the immunology of the mammary gland and find biomarkers to assess udder health. In conclusion, SCC is an indication of udder health or exposure of the udder to infectious agents or mechanical stress and should therefore always be considered a warning sign.

**Keywords:** goat; sheep; somatic cell count; mastitis; milk quality; physiology

## 1. Introduction

Dairy goat and sheep farming are important economically in many countries in the world. Their milk production is a valuable raw material for many milk products. The quality of these products is dependent upon the quality of raw milk used [1]. The somatic cell count (SCC) in SM is an important indicator of udder health and the raw milk quality and also reflects the animal's general health status [2–5], but the predictive values are better in ewes than in goats [6]. However, SCC plays a major role in protecting the milk epithelial cells (MEC) in the udder in general, which appears in the udder in response to invading pathogens. Polymorphonuclear neutrophils (PMNs), especially, invade in a higher number in the alveolar lumen. In addition, the SCCs in the SM are comprised of macrophages, lymphocytes, many cell fragments, and cytoplasmic particles/vesicles. PMNs were the major cell type in milk from healthy uninfected goats and constitute 40 to 80% of the SCC [7–12], whereas the macrophages (45–88%) are the major cell type in sheep's milk, with fewer PMNs (10–35%), lymphocytes (10–17%), and epithelial cells (2–3%) [7,10,13]. Mastitis (subclinical or clinical) increases the PMN content in sheep's milk but not in goat's

milk, in which they still represent the dominant type [9]. Thus, PMNs in goat's milk are the main cellular component in both healthy and in infected glands [14]. It is noteworthy that the function of PMNs in healthy mammary glands does not have the same effect as in mammary gland with mastitis [15]. The histopathological examination of goat udders with high SCCs demonstrated the absence of mastitis [16]. However, a high SCC value is normal in a healthy goat udder [8,17,18]. Various studies have reported that dairy goats and sheep have a different secretory system than other lactating animals. The apocrine system of goats and sheep produces cytoplasmic particles/vesicles (containing DNA) [10,19], and their milk can contain large numbers of cell fragments, resulting in exceeding the limit of SCCs, especially in the late stages of lactation [20]. The cytoplasmic particles/vesicles and cell fragments are counted "false positive" as somatic cells when samples are tested using automated counting methods. As a result, goat's and sheep's milk SCCs are higher than those found in other lactating animals. However, the concentrations of cytoplasmic particles/vesicles in goat milk are much higher than in ewe's milk [10]. The concentration of cytoplasmic particles/vesicles in goat's milk is about $150 \times 10^3$ cells/mL [20], whereas sheep's milk has about $15 \times 10^3$ cell/mL [7,21]. However, only 10% of SCCs are cytoplasmic particles/vesicles and cell fragments that are normally secreted with goat's milk [7]; the remaining 90% of the somatic cells are blood cells, which contribute to the immune defence of the mammary gland. In any case, good hygiene on the SM farms is a key to a low SCC in the milk produced and minimizes risks to guarantee an optimal consumption quality [3]. Furthermore, low levels of SCC in the milk are desirable to ensure good extraction of protein from raw milk, whereas high levels of SCC depress casein and other component levels in milk [3,22]. It has been shown that goats having high hemoglobin concentrations were characterized by a high milk yield and a lower SCC. In contrast, low hemoglobin concentrations (less than 5.6 mmol) resulted in higher cell counts [20]. Two main factors influence the SCC levels in sheep's and goat's milk—namely, physiological and environmental factors.

In the following, the physiological aspects of milk SCC in SM are summarized and discussed and could help to define the healthy and pathological physiological SCC levels in SM milk. Furthermore, it could help to gain deeper insight into the immunology of the mammary gland and find biomarkers to assess udder health in SM.

## 2. Physiological Level of SCC in Sheep Milk

So far, there is no clearly defined threshold value for physiological milk SCC in sheep that represents a healthy udder. However, the physiological level of SCC in ewe's milk is still under discussion. Several lines of evidence suggest that the physiological SCC levels in ewe's milk need to be limited in relation to mastitis [23]. Table 1 shows the physiological level of milk SCC in sheep's milk, and the mean value from all publications is $375 \times 10^3$ cells/mL. Previous studies have reported that the physiological level of milk SCC in dairy ewes ranged between $10 \times 10^3$ and $100 \times 10^3$ [24–27]. In a review by Tancin et al. [28], it was observed that SCC occurs at around $150 \times 10^3$ cells/mL in bulk milk tanks without mastitis in different breeds of sheep. Other researchers have pointed out that the upper threshold for milk SCC in the udder of a healthy ewe should be $250 \times 10^3$ cells/mL [29–31]. It was found in Manchega sheep that SCC $300 \times 10^3$ cells/mL was considered as the ideal value for the diagnosis of subclinical mastitis [32]. Zafalon et al. [33] found that the value of SCC above $400 \times 10^3$ cells/mL applies to the diagnosis of subclinical mastitis in herds. A similar cutoff value for the Santa Ines and Texel breeds in the lactation season $400 \times 10^3$ cells/mL in milk was reported by Kern et al. [34]. On the other hand, some studies clearly showed that the limit values for milk SCC in healthy udders are higher than previously announced and the limit value in different sheep breeds was $500 \times 10^3$ cells/mL [35–38]. Tvarožková et al. [39] also found that an SCC of more than $500 \times 10^3$ cells/mL of milk could be important for the detection of subclinical mastitis at half udder level in dairy sheep. Similar results reported that the threshold of SCC > $500 \times 10^3$ cells/mL was determined as an indicator of a change in milk

quality in ewes [40]. Likewise, it was clearly demonstrated that the negative milk samples from East Friesian ewes showed low somatic cell counts ($734 \pm 3153 \times 10^3$ cells/mL; $5.15 \pm 0.55$ log cells/mL) compared to the positive milk samples with higher somatic cell counts ($4432 \pm 6069 \times 10^3$ cells/mL; $5.97 \pm 0.96$ log cells/mL) [41]. In this study, it was shown that approximately 84% of the samples that tested negative had less than $500 \times 10^3$ cells/mL, and only 5% had more than $1000 \times 10^3$ cells/mL. Conversely, 32% of the positive samples had less than $500 \times 10^3$ cells/mL and 53% of the positive samples had more than $1000 \times 10^3$ cells/mL. Furthermore, Tancin et al. [28] reported that an SCC limit of $1000 \times 10^3$ cells/mL was established between healthy and mastitis in sheep. In any case, raw ewe's milk without pathogen bacteria had the lowest average SCC [42].

**Table 1.** Physiological levels of SCC in sheep's milk relative to breeds and geography.

| Location | Breed | Source of Milk Sample | SCC Level ($\times 10^3$ Cells/mL) | Authors |
|---|---|---|---|---|
| Spain | Menchega | Animal | 250 | [26] |
| Syria | Awassi | Right-udder half Left-udder half | 162 199 | [43] |
| Slovenia | Domestic highland, East Friesland a. Awassi | Udder half | 250 | [30] |
| Bulgaria | East Friesian $\times$ 1/4 East Friesian $\times$ Awassi | Animal | 1000 | [44] |
| Germany | East Friesian | Animal | 174 | [41] |
| Germany | East Friesian black and white | Udder half | 61 | [45] |
| Spain | Manchega sheep | * | 300 | [32] |
| Brazil | Corriedale and Texel | Animal | 317 | [46] |
| Italy | Sarda Sarda $\times$ Lacaune | Farm tank Farm tank | 206 171 | [47] |
| France | Lacaune Red-face Manech | Farm tank Farm tank | 500 1050 | [48] |
| Kosovo | Bar, Sha; Kos, Bal | Animal | 500 | [49] |
| Slovak Republic | Different races a. crossbred | Animal | 593 | [50] |
| Brazil | Lacaune | Animal | 1600 | [51] |
| Poland | Polish a. Polish Lowland | Udder half | 200 | [42] |
| Slovak Republic | Lacaune | Farm tank | 146 | [28] |
| Spain | Different races | Farm milk | 90 | [3] |
| Kashmir | Local breeds | Animal | 241 | [12] |
| Slovak Republic | Slovak dairy sheep a. Lacaune | Animal | 400 | [52] |
| Iraq | Local ewes | Animal | 39 | [53] |
| Italy | Sarda | Half-udder | 235 | [54] |
| Slovak Republic | Lacaune | Half udder | 200 | [55] |
| Greece | Different races a. crossbred | Farm tank | 501 | [56] |
| Mean | | | 375 | |

* Not specified.

### 3. Physiological Level of SCC in Goat's Milk

SCC in goat's milk is widely used for evaluating milk quality and as an indicator of udder health [5,57–59]. However, the SCC in goat's milk is generally higher than in sheep's and cow's milk [17], and SCC in milk as an indicator of IMI (Intramammary Infection) may be lower in goats than in sheep and cows since many physiological factors can increase SCC in the uninfected halves [11,17,60]. The average physiological level of SCC in goat milk was $764 \times 10^3$ cells/mL and fluctuated between $200 \times 10^3$ and $1500 \times 10^3$ cells/mL (Table 2). According to several authors, it is clear that higher SCC in goat's milk is bound with a high proportion of cell fragments. This means that the milk secretion in the goat, in contrast to the cow, leads to greater cell losses of the secretory glandular cells [8,19]. However, it has been confirmed that SCC is a relevant predictor of IMI and hygienic milk quality in goats [5], and SCC measured with a Delaval cell counter was strongly associated with bacterial growth in the half udder milk samples [61]. Rupp et al. [5] reported that goats with repeatedly healthy udders (80% of negative milk samples) had the lowest SCC values with an average of $277 \times 10^3$ cells/mL. According to the observations of Csanadi et al. [62], the average milk SCC of five different breeds of goat in Hungary was $664 \times 10^3$ cells/mL. Furthermore, milk SCC for goats free of IMI range between 270 and $2000 \times 10^3$ cells/mL [7]. The mean SCC for uninfected and infected halves with coagulated negative staphylococci were $272 \times 10^3$ and $932 \times 10^3$ cells/mL, respectively [22]. Contreras et al. [63] reported that a value of $500 \times 10^3$ cells/mL was a useful threshold to distinguish between infected and uninfected halves of the udder. The obtained results by Zeng and Escobar [16] demonstrated that 56% of the milking goats produced milk with $1000 \times 10^3$ SCC cells/mL; they concluded that healthy milk from healthy goats showing no signs of mastitis can contain up to $1000 \times 10^3$ SCC/mL. Values of $370 \times 10^3$ cells/mL milk [14] up to $1000 \times 10^3$ cells/mL milk are also given for the healthy goat udder [64]. A review by Koop et al. [65] suggested a cutoff value for SCC in goat's milk of $1500 \times 10^3$ cells/mL. According to Min et al. [66], mean SCC values in infected dairy goats ranged between 2000 and $4000 \times 10^3$ cells/mL, and they concluded that SCC in goat's milk does not strongly correlate with intramammary infection.

**Table 2.** Physiological levels of SCC in goat's milk relative to breeds and geography.

| Location | Breed | Source of Milk Sample | SCC Level $\times$ $10^3$ Cells/mL * | Author |
|---|---|---|---|---|
| Czech Republic | White shorthair | Animal | 600 | [18] |
| Czech Republic | White shorthair | Animal | 1422 | [67] |
| Greece | Different races a. crossbred | Farm tank | 770 | [56] |
| Italy | Alpine a. Saanen | Animal | 303 | [54] |
| Kashmir | Local races | Animal | 608 | [12] |
| Kosovo | Alpine a. native Red | Animal | 1000 | [68] |
| Spain | Different races | Farm tank | 660 | [3] |
| Spain | Local breed | Farm tank | 1200 | [7] |
| Sweden | Swedish landrace | Animal | 481 | [61] |
| Turkey | Saanen | Animal | 206 | [69] |
| USA | ** | Farm tank | 1150 *** | [70] |
| Mean | | | 764 | |

SCC *: This is geometric SCC; **: not specified; ***: by the fifth parity.

## 4. Considerations Relative to SCC Levels and Limits in Sheep's and Goat's Milk for Dairy

The legal SCC limit for bulk tank milk in the United States for sheep and goat is $750 \times 10^3$ cells/mL and $1000 \times 10^3$ cells/mL, respectively [7,62]. It is noteworthy that until now there was no milk SCC legal limit for sheep and goats in the European Union [62,67]. Corrales et al. [71] reported that the limit of $1500 \times 10^3$ cells/mL should be acceptable for goat's milk in the European Union. Currently, in the European Community the EU has yet to regulate values of SCC level in ewe's and goat's milk [3]. According to bulk tank milk, the SCC level can be established in both species as follows: 1. Good or acceptable ewe's or goat's milk when bulk tank milk SCC < $750 \times 10^3$ cells/mL; 2. Intermediate ewe's or goat's milk when bulk tank milk SCC is between 750 and $1500 \times 10^3$ cells/mL; and 3. Bad ewe's or goat's milk when bulk tank milk SCC > $1500 \times 10^3$ cells/mL [72,73].

## 5. Influence of Physiological Factors on SCC in Small Ruminants

It was observed that SCC level in SM can be affected by physiological factors such as milk fraction, breed, stage of lactation, parity, and type of birth [10,62,74–78]. An interesting aspect is that 48% of SCC variance can be attributed to physiological factors [79,80].

### 5.1. Effect of Milking Fraction

Usually, the first squirts before milking are the fraction used for bacteriological diagnosis and SCC determination. Several studies have shown that the first milk squirts have a similar SCC as in the main milking fraction. However, the first milk squirts have slightly lower SCC compared to main milking fraction [81], and the average values of SCC were $687 \times 10^3$ and $763 \times 10^3$ cells/mL, respectively [82]. A review by Martinez [83] found similar results, and the mean values of SCC were $998 \times 10^3$ and $1139 \times 10^3$ cells/mL in the first milk squirts and the main milking fraction, respectively. Kaskous [41] showed the opposite trend in East Friesian sheep in Germany, and the average SCC was $202.09 \times 10^3 \pm 471.02 \times 10^3$ cells/mL ($5.83 \pm 0.82$ log cells/mL) from the first milk squirts, whereas the average SCC reached to $174.27 \times 10^3 \pm 247.27 \times 10^3$ cells/mL ($5.44 \pm 0.62$ log cells/mL) from the total milk produced of each animal. In addition, research by Skapetas et al. [84] showed that SCC was lower in whole machine milk than in hand-stripped milk from Chios ewes.

### 5.2. Effect of Lactation Stage

Corrales et al. [85] found that SCC was very high at the end of lactation, and it is impossible to distinguish between infected and healthy mammary glands by SCC. It has been thought that the increase in SCC in the milk of SM with advanced lactation is due to a dilution effect [86–88]. This means that the SCC in milk is higher at the end of lactation due to the small amount of milk [27,89]. Paape and Capuco [19] emphasized this statement, and the SCC cannot increase in the late stage of lactation if the milk yield remains high until the end of lactation. Thus, the increase in SCC was associated with the progression of lactation in goats with or without a diagnosis of intramammary infection [86]. Studies by Kaskous [43] showed that normal SCC was higher in early and late stages of lactation (Figure 1).

Figure 1 shows that the available SCC values between 15 and 150 days of lactation were lower than $100 \times 10^3$ cells/mL. In other similar studies, SCC was lowest, averaging $200 \times 10^3$ cells/mL, at 15 days of lactation and reached a maximum of around $500 \times 10^3$ at 285 days [67]. Zafalon et al. [90] clearly show that the milk SCC values in sheep were higher at the end of lactation than in the second week of lactation, although the positive microorganisms in milk samples from half udders were higher in the second week of lactation (117/763, 15.3%) than at the end of lactation (86/694, 12.4%) (Table 3). One explanation for these contradictions is that the sheep produce less milk at the end of lactation, which increases the SCC value in the milk. This low milk yield at the end of lactation influences the lower incidence of subclinical mastitis. This is due to an increase

in macrophages and polymorphonuclear leukocytes in this phase, which are involved in udder defense during the dry period.

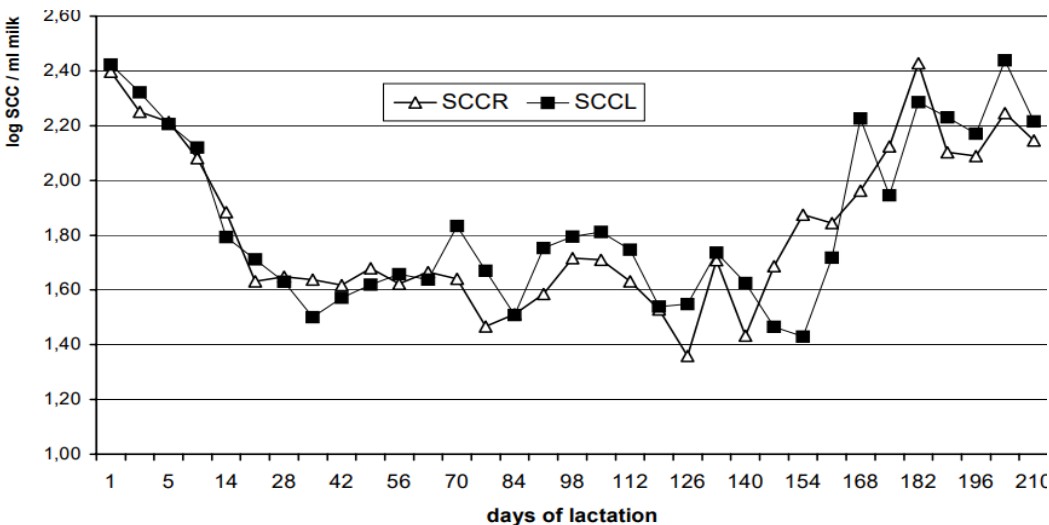

**Figure 1.** SCC level during lactation in Syrian Awassi ewes. SCCR: Somatic cell count in the right half of the udder; SCCL: Somatic cell count in the left half of the udder; log SCC: logarithmic somatic cell count. "Reprinted/adapted with permission from Ref. [43]. Copyright year 2021, copyright owner's name Kaskous". More details on "Copyright and Licensing" are available via the following link: https://www.mdpi.com/ethics#10.

**Table 3.** Negative and positive results of microorganism tests in samples of milk from mammary halves of different sheep breeds during the second week and at the end of lactation. "Reprinted/adapted with permission from Ref. [90] with some changes. Copyright year 2022, copyright owner's name: Kaskous". More details on "Copyright and Licensing" are available via the following link: https://www.mdpi.com/ethics#10.

| | Start of Lactation | | | End of Lactation | | |
|---|---|---|---|---|---|---|
| **Breeds** | **Number of Samples** | **Negative (% a)** | **Positive (% b)** | **Number of Samples** | **Negative (% a)** | **Positive (% b)** |
| Santa Ines | 402 | 76.6 | 23.4 | 343 | 81 | 19 |
| Texel | 120 | 94.2 | 5.8 | 116 | 95.7 | 4.3 |
| Ile de France | 120 | 90 | 10 | 104 | 91.3 | 8.7 |
| Dorper | 121 | 96.7 | 3.3 | 131 | 94.7 | 5.3 |
| Total | 763 | 84.7 | 15.3 | 694 | 87.6 | 12.4 |

[a]: Percentage negative of pathogens; [b]: Percentage positive of pathogens.

On the other hand, goat milk SCC is subject to greater physiological variability than cow milk SCC [91], and this variability is mainly related to lactation stage and goat breed [92]. Wilson et al. [86] have confirmed that more than 90% of SCC variations in goat milk may not be due to bacterial udder infections but to physiological factors such as lactation stage and month of the year or oestrus [93].

*5.3. Effect of Parity*

The influence of parity on the SCC seems to depend on the state of health of the udder and the pathogens involved [81]. A review by Paape et al. [67] observed an increase in SCC from the first to the fifth lactation. In general, in SM, increased SCC in milk was associated with higher parity [11,19]. The explanation for the fact that the content of somatic cells and

especially PMN in the milk increases with the number of lactations is not an age-related change but increased infection rates in older animals [94].

### 5.4. Effect of Type of Birth

Several reports have shown that type of birth (single, twin, or more birth) influences the SCC in the milk in SM [58,95]. However, the highest SCC values were achieved in animals with multiple births rather than in animals with a single birth [95]. On the contrary, some studies have shown that multiple births have no effect on SCC values [96]. In sheep and goats, some of them are known to have twins or more at birth. Such animals can produce more milk but also higher SCC, since udder health deteriorated in those sheep and goats that suckled two lambs instead of one. Threfore, suckling influenced later udder health during the milking period.

### 5.5. Effect of Milk Yield

The results from goats indicated that less productive animals without infection lead to higher SCC [83,86,96]. The converse is also correct. This means that goats producing >3 kg milk/day had the lowest SCC (<954 $\times$ 10$^3$ cells/mL) in the milk controls [95]. The results of several experiments support the hypothesis that the reduced milk production in goats and sheep is most likely due to the competent alveolar cells of the mammary gland. Therefore, the SCC in the milk increased significantly. This means that the alveoli have an impaired ability to secrete.

### 5.6. Effect of Breed

A review by Zafalon et al. [90] observed that the breed of sheep plays a very important role in SCC levels. The established limits of the SCC values for the diagnosis of subclinical mastitis were shown in different sheep breeds during the beginning and the end of lactation (Table 4). However, the SCC values in milk at the end of lactation were higher than those at the beginning of lactation.

**Table 4.** Cutoff values of SCC in ewe's milk of different breeds in Brazil. "Reprinted/adapted with permission from Ref. [90]. Copyright year 2022, copyright owner's name Kaskous". More details on "Copyright and Licensing" are available via the following link: https://www.mdpi.com/ethics#10.

| Breeds | Second Week of Lactation (SCC $\times$ 10$^3$ Cells/mL) | End of Lactation (SCC $\times$ 10$^3$ Cells/mL) |
|---|---|---|
| Santa Ines | 487 | 1171 |
| Texel | 419 | 802 |
| Ile de France | 781 | 554 |
| Dorper | 1062 | 1276 |

On the other hand, it was found that the SCC values in different dairy goat breeds (Alpine and Nubian) cannot confirm any significant differences [88], and the racial differences are due to differences in health, production levels, and management characteristics between them [97]. Similar results were reported by Csanadi et al. [62] in Hungary, and the SCCs of milk samples do not differ significantly depending on the genotype (Native, Saanen and Alpine $\times$ Saanen cross-bred). Based on the observations of Tancin et al. [28], the Lacaune breed of sheep was shown to have a higher percentage of mastitis milk (also higher SCC) compared to other breeds or crossbreeds (purebred Tsigai ewe, crossbred Slovak dairy ewe and crossbred Valachian $\times$ Lacanune ewes).

Finally, based on the observations of Paape et al. [7], it has been shown that physiological factors such as parity, stage of lactation, and milk yield did not have any clear influence on the SCC in milk, whereas intramammary infection has significantly increased the milk SCC in goats and sheep.

*5.7. Effect of Stress*

A new study in goats has clearly shown that the cumulative effect of various challenges imposed resulted in a change in hormone release and an increase in SCC in the milk produced as well as a reduction in milk yield [98]. In fact, stress in goats and sheep has been reported to increase SCC in the milk. Heat, vaccinations, dietary changes, and changes in milking routine are factors that lead to physiological stress. Since goats and sheep are very sensitive, the resulting decreases in milk yield under stress could explain the increase in SCC [98].

## 6. Increase of Somatic Cell Counts over Healthy Level in Small Ruminant

The main cause of increased SCC in milk of dairy ruminants is IMI [10,80,93,99–101]. Persson et al. [23] found a significantly higher SCC for udder halves with intramammary infection compared to udder halves without bacterial findings. As a result, high SCC has been suggested as the main reason for culling dairy sheep. But not every increase in the SCC in milk indicated an infection of the udder in SM [102]. Rupp et al. [5] reported that in lactating goats about 50% of the milk samples had a higher number of SCC ($1542 \times 10^3$ cells/mL) and no udder pathogens were detected, whereas positive udder pathogens with low SCC ($855 \times 10^3$ cells/mL) were detected in about 32.7% of the milk samples (Table 5). Tvarožková et al. [39] have shown clear results that SCC $\geq 500 \times 10^3$ cells/mL were detected in 92.5% bacteriologically positive milk samples. This means that the presence of pathogens increased the SCC in the milk significantly ($p < 0.001$) compared to samples that were free of pathogens. This resembles the results of Olechnowicz and Jaskowski [22], which showed that total bacterial count is significantly correlated with the number of somatic cells in bulk milk. Likewise, results from 155 French herds showed that annual geometric means of $750 \times 10^3$, $1000 \times 10^3$, and $1500 \times 10^3$ SCC/mL corresponded to $30 \pm 12\%$, $39 \pm 8\%$, and $51 \pm 8\%$ of infected goat udders, respectively [103,104].

**Table 5.** Milk bacteriological results for the high-and low SCC in milk samples from healthy goats or goats with nonacute mastitis, with some changes. "Reprinted/adapted with permission from Ref. [5] with some changes. Copyright year 2022, copyright owner's name Kaskous". More details on "Copyright and Licensing" are available via the following link: https://www.mdpi.com/ethics#10.

| Item | Total [a] | | Low-SCC $855 \times 10^3$ | | High-SCC $1542 \times 10^3$ | |
|---|---|---|---|---|---|---|
| | N | % | N | % | N | % |
| Negative milk samples | 1547 | 57.55 | 757 | 67.3 | 790 | 50.5 |
| Positive milk samples | 1141 | 42.45 | 367 | 32.7 | 774 | 49.5 |
| -Staphylococci (total) | 922 | 34.3 | 281 | 25 | 641 | 41 |
| -Streptococci | 75 | 2.8 | 27 | 2.4 | 48 | 3.1 |
| -Bacillus | 102 | 3.8 | 47 | 4.2 | 55 | 3.5 |
| -Micrococci | 22 | 0.8 | 8 | 0.7 | 14 | 0.9 |
| -Others [b] | 20 | 0.7 | 4 | 0.4 | 16 | 1.0 |

[a] Results were from 2688 milk samples from 9 sampling time points per udder half during first lactation. [b] Others: Acinetobacter, Aerococcus, Aspergillus, Corynebacterium, Escherichia coli, Pseudomonas and yeast.

## 7. Influence of the Milking Procedures on the SCC Level in SM

Proper milking procedures are essential to minimizing the risk of a bacterial infection causing mastitis and to reducing the SCC in the milk produced [105]. Therefore, milk SCC level is influenced by machine milking [105–108]. The prevention of udder diseases is mainly based on milking machine management, hygiene, annual milking machine checks, and optimization of milking technology [6]. It is essential that the milking system meet the physiological requirements of sheep and goats in order to increase milk yield and achieve better milk quality. Thus, the udder remains healthy [105,109]. Besides the liner, there

are three operating parameters that regulate mechanical milking: vacuum level, pulsation rate, and pulsation ratios. The milking system needs to provide a stable vacuum, adequate pulsation, and gentle milking action. High vacuum levels (>42 kPa) are often used during machine milking in dairy ewes and goats [110]. However, the use of 44 kPa in the goat milking machine showed an increase in the tissue thickness of the teats over 5%, and the average conductivity of the milk tended to increase [109]. Consequently, higher SCC can be found in the produced milk in SM. Several reports have shown that the vacuum required to open the teat canals in the sheep and goats is between 25 and 35 kPa [111,112]. On the other hand, sheep and goats store the milk in their gland cistern more than in the milk-producing alveoli's udder [113]. However, cisternal milk fraction of SM can range from 50% in sheep [114] to 85% in goats [115]. This, along with the smaller teat size, makes a faster pulsation an acceptable option. In recent years many milking machines for sheep and goats have been developed. However, many farmers are suffering from the performance of these milking machines, as they do not empty the udder completely and many udders will become diseased as well as the SCC increased in the milk produced [84,116,117]. The cause of this problem is the absence of proper milking machines that adapt to the physiology of sheep's and goats' udders. For this reason, Siliconform from Germany has developed a new milking machine for sheep and goats, which includes teat cups with an integrated air inlet (Figure 2).

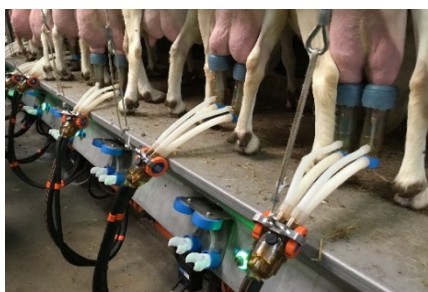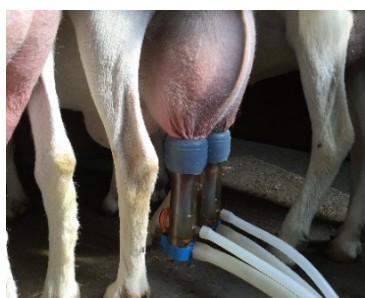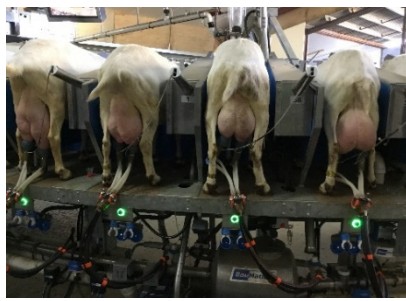

**Figure 2.** Goats were milked with the new milking machines from Siliconform-Germany (pictures from Kaskous, 2020). Permission has been obtained and there is no copyright issue.

For SM, it is important during machine milking to monitor mammary gland health through diagnostic testing and teat dipping after milking is complete [2,6]. However, improving hygienic conditions during milking is a key to assessing the microbiological quality of the milk [22]. It is noteworthy that the correlation coefficients between SCC and morphological traits of the udder suggest that some of the morphological traits need to be taken into consideration in the evaluation of the suitability of SM for machine milking [118–120]. It has been estimated that ewes with a more horizontal teat position and larger teats had higher SCC, since they are more prone to develop subclinical mastitis [121]. Due to the above observation, it seems necessary to show the course and consequences of machine milking in SM on milk quality and health of the teat end [22]. In addition, the type of milking had a significant impact on bulk tank SCC in ewes [22].

Bulk tank SCC is also affected by the frequency of milking in dairy goats [122–124]. To improve the udder health status of SM, it is necessary to ensure hygienic conditions of animal maintenance and optimization of milking machine standards and parlour systems.

## 8. Impact of High Somatic Cell Counts on Milk Quality and Processing

The determination of SCC in sheep's and goat's milk is very important for milk processors, because SCC with other factors plays a big role in determining safety and hygienic quality of the final product [3,62]. A review by Podhorecka et al. [18] observed that low SCC values may significantly affect the technological properties of goat's milk, and SCC should therefore be routinely screened by dairy manufacturers to assure the consumer of high end-product quality. The study by Giaccone et al. [125] clearly showed that an

influence of a higher milk SCC of the Valle del Belice sheep breed influences the properties of cheese production. The same authors found that the fat, protein, casein, and lactose contents in the milk of the higher cell number group ($>1000 \times 10^3$ SCC/mL) were lower than that of the lower cell number group ($<1000 \times 10^3$ SCC/mL) (Table 6).

**Table 6.** Effect of somatic cell count on milk quality in sheep, "Reprinted/adapted with permission from Ref. [125] with some changes. Copyright year 2022, copyright owner's name Kaskous". More details on "Copyright and Licensing" are available via the following link: https://www.mdpi.com/ethics#10.

| Parameters | High Level of SCC ($>1000 \times 10^3$) | Low Level of SCC ($<1000 \times 10^3$) |
|---|---|---|
| Somatic cell count ($\log_{10}$) | 6.40 [A] | 5.56 [B] |
| Fat (%) | 6.29 [A] | 6.92 [B] |
| Protein (%) | 5.27 | 5.32 |
| Casein (%) | 4.25 | 4.39 |
| Lactose (%) | 4.38 [a] | 4.71 [b] |
| Whey protein (%) | 1.08 | 1.03 |
| Urea (mg/dl) | 31.69 [A] | 33.07 [B] |
| pH | 6.79 [A] | 6.68 [B] |
| Calcium (g/l) | 1.89 | 1.93 |
| Phosphorus (g/l) | 1.42 | 1.35 |

Within row, different letters are significant at $p < 0.05$ (small letters) and at $p < 0.01$ (capital letters).

Furthermore, the casein fractions ($\alpha$S1-casein and $\beta$-casein) and fat content in fresh feta cheese were significantly lower when the bulk tank of SCC was high, whereas pH and fatty acid content were increased [22]. It is known that an increase of SCC causes a decrease in milk yield and affects milk composition [16,28,126–128], which leads to reduced cheese-making potential [129]. Moreover, it has been estimated that milk with three levels of SCC content in sheep showed differences ($p < 0.01$) in whey protein, lactose, pH, and total Na content. Furthermore, protein recovery rate was higher in cheese from low SCC milk, whereas adjusted cheese yield did not show significant differences [130]. Consequently, Csanadi et al. [62] noted that goat's milk having very high SCC is not suitable for making any goat milk products.

On the contrary, Chen et al. [131] found that milk composition did not change when milk SCC varied from $214 \times 10^3$ to $1450 \times 10^3$ cells/mL in Alpine goats without evidence of clinical mastitis, and no significant differences in the yield of semisoft goat cheeses were detected. However, total sensory scores and body and texture scores (hardness, springiness) for cheeses made from the high SCC milk were lower than those for cheeses made from the low and medium SCC milk. Moreover, it is also noted that individual and total free fatty acid (FFA) significantly increased ripening, regardless of the SCC levels.

## 9. Potential Use of Differential Cell Count in Milk of Small Ruminants

Since milk leukocytes comprise different cell types, each with specific roles in the immune defense of the mammary gland [132], it could be valuable to perform a differential cell count (DCC), which has already proven beneficial in dairy cows [133]. Today, there are even automated cytometers available to determine the main populations of immune cells in cow's milk [134,135]. Moreover, using fluorochrome-conjugated antibody to capture a particular cluster of differentiation (CD) molecule on the cell's surface, it is possible to further differentiate subpopulations of immune cells based on their specific wavelength [136]. Winnicka et al. [137], for example, used this method to analyze blood and milk leukocytes of healthy goats to document the percentages of different lymphocyte populations over

the course of lactation. The authors showed that the percentage of T helper cells (CD4+) increased during early lactation until day 14 and decreased in mid-lactation, whereas the amount of cytotoxic T cells (CD8+) increased until day 21 and then remained stable. For dairy cows, the ratio of CD4+ to CD8+ was proposed as a biomarker of low mastitis resistance, with a threshold below 1 [138]. Tatarczuch et al. [139] examined secretions of involuting udders of sheep and found CD8+ T cells to be the predominant lymphocyte population during early involution. In another flow cytometric study, the authors additionally determined the viability of several subpopulations in caprine milk [140]. Blagitz et al. [141] observed a reduced viability of PMN in goat milk samples with a low SCC and suggested a higher susceptibility to intramammary infections in such cases. For dairy cows, several biomarkers have already been introduced to diagnose mastitis (e. g., the logarithmic ratio of polymorphonuclear neutrophilic leukocytes to lymphocytes [142] or the expression of the inflammatory marker β-integrin CD11b on milk leukocytes [143]). In a different approach, Farschtschi et al. [144] used three different vaccines as systemic immune stimuli and could demonstrate an influence on both blood and milk DCCs. Further research in DCC could help to gain a deeper insight into the immunology of the mammary gland of SM and find biomarkers to assess udder health [10] or even evaluate the systemic immune status, as proposed for dairy cows [144].

## 10. Conclusions

Intramammary infection is the main cause of increased SCC in milk of lactating animals. However, physiological factors such as lactating stage, lactation number, type of birth, milk yield, and breed have also an influence on the SCC in sheep and goats. It is also noted that the physiological SCC levels in milk remain low when housing, feeding, and milking are carried out under ideal conditions. Importantly, the milking system must meet the physiological requirements of sheep and goats in order to increase milk yield, achieve better milk quality, and maintain udder health. So far, there are currently no legal limits for somatic cells in goat's and sheep's milk. However, a maximal limit of $750 \times 10^3$ cells/mL for sheep's milk and $1000 \times 10^3$ cells/mL for goat's milk has been set in the USA. In relation to this study, the threshold in the US is high and our recommended SCC values in sheep's and goat's milk are $500 \times 10^3$ cells/mL and $750 \times 10^3$ cells/mL, respectively. It is noteworthy that this proposed high SCC threshold in goat's milk is due to the high concentration of cytoplasmic particles/vesicles.

**Author Contributions:** S.K., conceptualization and writing the original manuscript, except point number 9, which was written by S.F.; M.W.P., revision of some content and language improvement of the original manuscript. All authors have read and agreed to the published version of the manuscript.

**Funding:** There is no additional funding for this research other than salary.

**Institutional Review Board Statement:** Not applicable.

**Informed Consent Statement:** Not applicable.

**Data Availability Statement:** All data used to support the findings of this study are included within.

**Conflicts of Interest:** Shehadeh Kaskous: As head of the research and development department, I am an employer at Siliconfrom. No company materials or equipment were used as the paper is a review. There was no financial support other than my salary. Shehadeh Kaskous was the senior author involved in the design of the manuscript.

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
