# Peer review of "Physiological Aspects of Milk Somatic Cell Count in Small Ruminants—A Review"

_2624-862X, doi:10.3390/dairy4010002_

Round 1

Reviewer 1 Report

I noted my comments below.

Manuscript ID dairy-2017415

- Lines 86-88 Table (1) shows the physiological level of milk SCC in sheep milk, and the mean value from all publications is 375 x 103 cells/ml.

I consider uncorrected to report and comment on the average value as obtained in table 1. This mean value certainly derives from values that consider different breeds and types of sampling…This consideration is also supported by Authors (lines 162-164)

- Lines 103-105 Likewise, it was clearly demonstrated that the negative milk samples from East Friesian ewes showed low somatic cell counts (734±3153 x 103 cells/ml; 5.15±0.55 log cells/ml) compared to the positive milk samples with higher somatic cell counts (4432±6069 cells/ml; 5.97±0.96 log cells/ml)

The SD reported is very huge (734±3153; 4432±6069); please comment these reported data.

- Lines 120-121 The average physiological level of SCC in goat milk was 764 x 103cells/ml and fluctuated between 200 x 103 and 1500 x 103 cells/ml (Table 2).

For table 2 I point out the same comment reported previously for table 1.

- Lines 127-139.

The Authors report published data, however my suggestion is to comment more deeply on these cited papers.

- Lines 200-202 One explanation for these contradictions is that the sheep produce less milk at the end of lactation, which increases the SCC value in the milk. This low milk yield at the end of lactation influences the lower incidence of subclinical mastitis.

I don't fully understand the Authors’ comments.

- Tables 3, 4, 5 and 6.

All these tables are based on data published in a single paper. The same data presented in the tables are also reported in the text. These tables add few information, especially for a review.

- Lines 225 5.5. Effect of milk yield on SCC

This paragraph is not sufficiently commented and addressed in an inexhaustive way.

Author Response

Thank you for your comments which have raised the level of the manuscript.

Lines 86-88 Table (1) shows the physiological level of milk SCC in sheep milk, and the mean value from all publications is 375 x 103 cells/ml. I consider uncorrected to report and comment on the average value as obtained in table 1. This mean value certainly derives from values that consider different breeds and types of sampling…This consideration is also supported by Authors (lines 162-164).

Answer: Yes, breed of sheep and type of milk sample certainly play a role, but just as with cows, despite different breeds and milk samples, there is an official value.

Lines 103-105 Likewise, it was clearly demonstrated that the negative milk samples from East Friesian ewes showed low somatic cell counts (734±3153 x 103 cells/ml; 5.15±0.55 log cells/ml) compared to the positive milk samples with higher somatic cell counts (4432±6069 cells/ml; 5.97±0.96 log cells/ml). The SD reported is very huge (734±3153; 4432±6069); please comment these reported data.

Answer: In this study, it was shown that approximately 84% of the samples that tested negative had less than 500 x 103 cells/mL and only 5% had more than 1000 x 103 cells. Conversely, 32% of the positive samples had less than 500 x 103 cells/mL and 53% of the positive samples had more than 1000 x 103 cells/mL.

Lines 120-121 The average physiological level of SCC in goat milk was 764 x 103cells/ml and fluctuated between 200 x 103 and 1500 x 103 cells/ml (Table 2). For table 2 I point out the same comment reported previously for table 1.

Answer: Yes, breed of goat and type of milk sample certainly play a role, but just as with cows, despite different breeds and milk samples, there is an official value.

-Lines 127-139. The Authors report published data; however, my suggestion is to comment more deeply on these cited papers.

Answer: Yes, we can do that, but the paper will be very long and contain a lot of information.

Lines 200-202 One explanation for these contradictions is that the sheep produce less milk at the end of lactation, which increases the SCC value in the milk. This low milk yield at the end of lactation influences the lower incidence of subclinical mastitis. I don't fully understand the Authors’ comments.

Answer: Yes, you are right, I explained it further in the text. This is due to an increase in macrophages and polymorphonuclear leukocytes in this phase, which are involved in udder defense during the dry period.

Tables 3, 4, 5 and 6.: All these tables are based on data published in a single paper. The same data presented in the tables are also reported in the text. These tables add few information, especially for a review.

Answer: I see your point, but for publications and reviews its good practice to include tables and figures for an instant overview.

Lines 225 5.5. Effect of milk yield on SCC: This paragraph is not sufficiently commented and addressed in an inexhaustive way.

Answer: Yes, you are right. I explained it further in the text. The results of several experiments support the hypothesis that the reduced milk production in goats and sheep is most likely due to the competent alveolar cells of the mammary gland. Therefore, the SCC in the milk increased significantly. This means that the alveoli have an impaired ability to secrete.    

Reviewer 2 Report

The current manuscript effectively reviewed the relevant literature about the physiological levels of somatic cell count (SCC) in small ruminants. The authors discussed different levels of SCC in small ruminants with respect to animal physiology and management practices. There are limited reviews on SCC in small ruminants that makes this work important for the readers. Overall, the manuscript was easy to follow and well-structured. There are quite a few grammatical errors which need Academic English Editing. My major suggestion is to perform the meta-analysis to define the acceptable limits of SCC in small ruminants. That would increase the importance of this review to a much larger extent.   

Some minor suggestions.

Title:

Use the plural word “small ruminants” instead of “small ruminant”.

Abstract:

The abstract in current form is > 325 words. Please limit it to 200 words. Follow the style of your conclusion section. It will help to concise the abstract.

Line 30-31: The manuscript had just few studies on bulk tank SCC. It would be great to avoid describing findings from these studies in the abstract.

Table 1 and Table 2:

Without the meta-analysis, it could be misleading to present the mean values of the SCC.  I would strongly recommend that the authors should perform meta-analysis to establish threshold levels of SCC.

Line 307: There is no need to have this figure. Also, please avoid using the name of the company while describing the milking machine characteristics.  

Author Response

Thank you for your comments which have raised the level of the manuscript.

The current manuscript effectively reviewed the relevant literature about the physiological levels of somatic cell count (SCC) in small ruminants. The authors discussed different levels of SCC in small ruminants with respect to animal physiology and management practices. There are limited reviews on SCC in small ruminants that makes this work important for the readers. Overall, the manuscript was easy to follow and well-structured. There are quite a few grammatical errors which need Academic English Editing. My major suggestion is to perform the meta-analysis to define the acceptable limits of SCC in small ruminants. That would increase the importance of this review to a much larger extent.  

Answer: We have attempted to carefully read the manuscript again to correct the English error. Because a meta-analysis is a good idea. We want to do a meta-analysis in other scientific papers.

Title: Use the plural word “small ruminants” instead of “small ruminant”.

 Answer: Thank you for this comment, we have corrected in the text.

Abstract: The abstract in current form is > 325 words. Please limit it to 200 words. Follow the style of your conclusion section. It will help to concise the abstract.

Answer: Thanks for your comment. We deliberately wrote the abstract in this way because many scientists only want to or can read the abstract. Therefore, we have written a comprehensive abstract.

Line 30-31: The manuscript had just few studies on bulk tank SCC. It would be great to avoid describing findings from these studies in the abstract.

Answer: Thank you for your comment and we have taken your comment into account in the text.

Table 1 and Table 2: Without the meta-analysis, it could be misleading to present the mean values of the SCC.  I would strongly recommend that the authors should perform meta-analysis to establish threshold levels of SCC.

Answer: Thanks for your comment. We wrote above that we want to do a meta-analysis on this topic in other scientific papers.

Line 307: There is no need to have this figure. Also, please avoid using the name of the company while describing the milking machine characteristics.

Answer: Thanks for your comment. It has always been recommended to show pictures and tables to give the reader an immediate overview. The company's name was written intentionally to inform scientists about new technologies. 

Reviewer 3 Report

The authors review the physiological aspects of milk somatic cell count (SCC) in small ruminants, including goat and sheep. The manuscript contains some novel contents. Some queries have arisen during my review, which I list below for the authors to consider.

1. Please revise the ml to mL throughout the manuscript.

2. Please remove the “to these cells” in L44.

3. Please revise the and  to comma between lymphocytes and many cell fragmentsin in L45.

4. Please add the cells/mL between 1000×103 and respectively in L151.

5. Please add the the between first milk squirts and and main milking fraction in L173.

6. Please analyse the reasons for paragraphs 5.3, 5.4, 5.5 and 5.7.

Author Response

Thank you for your comments which have raised the level of the manuscript.

The authors review the physiological aspects of milk somatic cell count (SCC) in small ruminants, including goat and sheep. The manuscript contains some novel contents. Some queries have arisen during my review, which I list below for the authors to consider.

Answer: Thank you for your comment.

  1. Please revise the “ml” to “mL” throughout the manuscript

Answer: Thanks for the comment, Itʹs done.

  1. Please remove the “to these cells” in L44.

Answer: It's done

  1. Please revise the “and” to “comma” between lymphocytes and many cell fragments in L45.

Answer: It's done.

  1. Please add the “cells/mL” between 1000×103and respectively in L151

Answer: It's done.

  1. Please add the “the” between “first milk squirts and” and “main milking fraction” in L173.

Answer: It's done.

  1. Please analyse the reasons for paragraphs 5.3, 5.4, 5.5 and 5.7

Answer: It's done. You can see the information in the text.

Round 2

Reviewer 2 Report

The authors justified their responses to the technical comments. An academic English editing may still improve the quality of the text. Some minor suggestions are listed below. 

Line 1: The title still reads as “small ruminant” instead of “small ruminants” as suggested previously.

Line 11: The text “in majority in goat and sheep” does not add anything to the sentence. Delete it.

Line 13: Use “ruminants” to convey the idea effectively and follow this comment throughout the manuscript where needed.

Line 13: The use of “following” in the sentence may be replaced with appropriate word.

Line 29: Was it “herd”?

Line 224: It is a very strong statement and needs a reference.

Line 268-272: Do these statements have the reference Vasconcelos et al. [97]? If not, then add the appropriate references.

Line 324: Please add the source of figure 2 in the references. I could not trace it.

Author Response

Thank you for all comments. We took them all and marked them in red in the manuscript.

Line 1: The title still reads as “small ruminant” instead of “small ruminants” as suggested previously.

Answer: It is done.

Line 11: The text “in majority in goat and sheep” does not add anything to the sentence. Delete it.

Answer: It is done.

Line 13: Use “ruminants” to convey the idea effectively and follow this comment throughout the manuscript where needed.

Answer: It is done.

Line 13: The use of “following” in the sentence may be replaced with appropriate word.

Answer: It is done.

Line 29: Was it “herd”?

Answer: It is done.

Line 224: It is a very strong statement and needs a reference.

Answer: It is done.

Line 268-272: Do these statements have the reference Vasconcelos et al. [97]? If not, then add the appropriate references.

Answer: It is done.

Line 324: Please add the source of figure 2 in the references. I could not trace it.

Answer: These images in figure 2 have not been published.